# Study of the Dynamic Recrystallization Process of the Inconel625 Alloy at a High Strain Rate

**DOI:** 10.3390/ma12030510

**Published:** 2019-02-08

**Authors:** Zhi Jia, Zexi Gao, Jinjin Ji, Dexue Liu, Tingbiao Guo, Yutian Ding

**Affiliations:** 1State Key Laboratory of Advanced Processing and Recycling of Nonferrous Metals, Lanzhou University of Technology, Lanzhou 730050, China; dxliu@lut.cn (D.L.); guotb@lut.cn (T.G.); dingyt@lut.edu.cn (Y.D.); 2School of Material Science and Engineering, Lanzhou University of Technology, Lanzhou 730050, China; gaozexi0525@163.com; 3School of Materials Engineering, Lanzhou Institute of Technology, Lanzhou 730050, China; 617599947@qq.cpm

**Keywords:** deformation behavior, microstructure, dynamic recrystallization, texture, Inconel625 alloy

## Abstract

High-temperature compression and electron backscatter diffraction (EBSD) techniques were used in a systematic investigation of the dynamic recrystallization (DRX) behavior and texture evolution of the Inconel625 alloy. The true stress–true strain curves and the constitutive equation of Inconel625 were obtained at temperatures ranging from 900 to 1200 °C and strain rates of 10, 1, 0.1, and 0.01 s^−1^. The adiabatic heating effect was observed during the hot compression process. At a high strain rate, as the temperature increased, the grains initially refined and then grew, and the proportion of high-angle grain boundaries increased. The volume fraction of the dynamic recrystallization increased. Most of the grains were randomly distributed and the proportion of recrystallized texture components first increased and then decreased. Complete dynamic recrystallization occurred at 1100 °C, where the recrystallized volume fraction and the random distribution ratios of grains reached a maximum. This study indicated that the dynamic recrystallization mechanism of the Inconel625 alloy at a high strain rate included continuous dynamic recrystallization with subgrain merging and rotation, and discontinuous dynamic recrystallization with bulging grain boundary induced by twinning. The latter mechanism was less dominant.

## 1. Introduction

Due to its strength at high temperatures, structural stability, oxidation resistance, and resistance to hot corrosion, the Inconel625 alloy is widely applied in aerospace, nuclear power, shipbuilding, and other industrial fields. At present, the hot extrusion of pipes and bars are used in the processing and forming of nickel-based superalloys in various fields. However, the Inconel625 alloy has a narrow range of thermoforming parameters, a complex composition, and a high deformation resistance. This results in poor processing stability, where uneven stress and temperature distribution, and difficulties in controlling the extrusion process result in the occurrence of bursting—especially for extrusion at a high deformation rate and in the formation of thin-walled tubes. In order to solve these problems, it is imperative to investigate the dynamic recrystallization behavior and microstructure evolution of the Inconel625 alloy. At present, research on superalloys has focused on a variety of aspects. Some studies have investigated the microstructure analysis and hot deformation behavior of nickel-based superalloys [1,2,3,4,5,6,7,8,9,10,11,12,13]. In these studies, optical microscopy (OM), transmission electron microscopy (TEM), electron backscatter diffraction (EBSD), and other analytical techniques were used to study the grain morphology, grain boundary evolution, and dynamic recrystallization nucleation mechanism of nickel-based superalloys under different hot deformation conditions. The hot deformation parameters have significant influences on the grain microstructure. The fraction of low-angle grain boundaries decreases with the increase of temperature and the decrease of strain rate. The bulging of initial grain boundaries is the main nucleation mechanism of dynamic recrystallization (DRX). Other studies have focused on the work-hardening behavior of nickel-based superalloys during hot deformation, and the constitutive model based on the stress–strain relationship [14,15,16,17,18,19,20,21,22,23,24,25]. The hot deformation behaviors of the Ni-based superalloy are investigated by tensile and compressional tests over a wide range of strain rates and deformation temperatures. The results show that the flow stress is sensitive to strain, strain rate, and deformation temperature. In addition, studies have investigated the dynamic recrystallization behavior of superalloys during hot deformation [26,27,28,29,30,31,32,33]. Dynamic recrystallization occurs during the hot-working of a metallic material with low-to-medium stacking–fault energy. The macroscopic mechanical behavior during hot-working is largely affected by the microstructure evolution due to DRX. The results indicate that both continuous dynamic recrystallization and discontinuous dynamic recrystallization take place during hot deformation. Discontinuous dynamic recrystallization (characterized by grain boundary bulging) and continuous dynamic recrystallization (characterized by subgrain rotation) play different dominant roles for the studied superalloys under different hot deformation conditions.

Although there are extensive studies on various aspects of nickel-based superalloys, literature on the hot deformation behavior and microstructure analysis at high temperatures and high strain rates is still very limited. Therefore, this study conducted a detailed analysis of the Inconel625 alloy at a high strain rate at different temperatures.

## 2. Materials and Experimental Methods

The Inconel625 alloy was a solid, solution-strengthened nickel-based deformation superalloy. The initial specimens with a size of Φ8 mm × 12 mm, and the compressed sample are shown in Figure 1a. The chemical compositions were measured by Electron Microprobe (EPMA1600, Lanzhou, China) as shown in Table 1. The original microstructure of the Inconel625 alloy is shown in Figure 1b; the structure was extremely uneven. The hot compression tests were performed using a Gleeble 3800 (Gleeble, Shanghai, China) instrument at 900–1200 °C and strain rates in the range of 0.01–10 s^−1^. Prior to loading, all specimens were heated to the experimental temperatures at a heating rate of 5 °C/s, and then held for 5 min. The height reduction of the specimens was 50%, as shown in Figure 1a. After the hot compression tests, the cross section of the sample center was determined using a scanning electron microscope (SEM) with an EBSD system (6700F, Lanzhou, China). For the observations, alloy flakes were machined from the hot compression specimens. They were ground to a thickness of 1–3 mm, and disks 3 mm in diameter were punched out.

## 3. Results 

### 3.1. The True Stress–Strain Curves and Constitutive Model

The true stress–strain curves shown in Figure 2 depict the effect of different temperatures and strain rates on the Inconel625 alloy. It was evident that the temperature and strain rate had the most important effect on the flow behavior of the Inconel625 alloy. The rheological curve can be roughly divided into three stages. During the work-hardening stage the stress increased rapidly with the strain, and then gradually decreased after reaching a peak. The main reason for this stress change was that a large number of dislocations accumulated in the initial stage of high-temperature compression. This resulted in an increase in the dislocation density, and in limited movement, which limited the deformation of the material. The second stage was the dynamic softening stage, which corresponded to a decrease in the stress. At this time, dynamic softening and DRX occurred successively due to the increase in the dislocation density up to a certain critical value. As a result, the dynamic softening effect was stronger than the work-hardening effect. The last stage was the steady–state phase, which corresponded to stabilization of the stress. With continuous deformation, some dynamically recrystallized grains grew and increased the work-hardening effect. Ultimately, a balance was achieved between the work-hardening effect and the dynamic softening effect. It was observed that all the stress–strain curves were jagged in the second and third stages, which also indicated that the work-hardening and dynamic softening effects alternated during these two stages. 

The peak stress result (Figure 2e) indicated that the temperature and strain rate had different effects on peak stress. At the same strain rate, the true stress decreased with increasing temperature. This occurred because, at high temperatures, the dynamic recrystallization process was accelerated and the softening effect was enhanced, resulting in a decrease in the true stress. At the same temperature, the true stress was higher at a higher strain rate. This was because at high strain rates, the deformation time was shorter and the dynamic recrystallization was incomplete, resulting in a reduced softening effect and an increase in the true stress.

An adiabatic heating effect was observed during the hot compression process, as shown in Figure 3. At a strain rate lower than 0.1 s^−1^, the adiabatic heating effect was particularly weak because the deformation process was relatively slow, and there was sufficient time to dissipate the heat generated by the deformation. However, when the strain rate was greater than 1 s^−1^, the adiabatic heating effect was more pronounced. Moreover, as the temperature increased, the adiabatic heating effect was weaker. This was because as the temperature increased, the dynamic recrystallization nucleation was accelerated and required a large amount of heat, which was absorbed by the material during deformation. 

According to the previous studies, the Arrhenius model has an advantage in determining the rheological behavior at elevated deformation temperatures, which is expressed as:(1)$$\dot{\varepsilon }=A{\left[\sinh\left(\alpha \sigma \right)\right]}^{n}\exp\left(-\frac{Q}{RT}\right),$$
where *A*, α, and *n* are material constants, *Q* is the activation energy of deformation, *T* is the absolute temperature, *R* is the universal gas constant (8.314 J∙mol^−1^∙K^−1^), and $\dot{\varepsilon }$ is the strain rate. Based on the peak stress from the hot compression test at different strain rates, the relationships of Inσ −$\;\mathrm{In}\;\dot{\varepsilon }$, $\sigma -\;\mathrm{In}\;\dot{\varepsilon }$, In[sinh(ασ)] − $\mathrm{In}\;\dot{\varepsilon }$ and In[sinh(ασ)] − T^−1^ can be determined, as shown in Figure 4. The value of α, *A*, and *Q* can be obtained through the relationships of lines shown in Figure 5. Thus, the constitutive equation of the Inconel625 alloy was obtained from the following: (2)$$\varepsilon =5.816\times 10^{17}{\left[\sin h\left(0.0033\sigma \right)\right]}^{4.873}\times \exp\left(-\frac{473.713}{RT}\right).$$

### 3.2. The Dynamic Recrystallization Behavior of the Inconel625 Alloy

Figure 6a,b,d marks the diameter between 0 and 10 μm grains with different colors at different temperatures. Figure 6c shows the diameter between 0 and 5 μm grains. Figure 6e,f depicts the average grain size and average grain aspect ratio. It was evident that the grain size was very sensitive to the temperature at a high strain rate of 50% deformation. At 900 °C, some grain boundaries were bent into a stepped pattern due to the stress concentration, as shown in Figure 12a. These stepped grain boundaries hindered the movement of dislocations, resulting in an increase in the dislocation density. The dynamic recrystallization nucleation preferentially adhered adjacent to these grain boundaries, as shown in Figure 6a. As shown in Figure 6b, as the deformation temperature increased, the 0–10 μm grains increased rapidly. The initial large grains gradually disappeared and were replaced by dynamically recrystallized grains, and the average grain diameter rapidly decreased to 3.86 μm. Because the newly formed dynamic recrystallized grains were free of distortion, and the dislocation density of the original grains was high, the imbalance caused the growth of the grains, increasing the average grain size to 7.62 μm at 1100 °C. As shown in Figure 6f, the aspect ratio reached 1.65, indicating that the fully equiaxed grains were obtained. At a temperature of 1200 °C (Figure 6d), the grains in the size range of 0–10 μm had almost disappeared, owing to the high temperature and the rapid deformation rate. The grain boundary migration rate was particularly fast, resulting in grain growth by mutual annexation, with an average grain diameter of 20.87 μm. It is generally accepted that the growth of the grains is realized by the motion of the grain boundary, which is not only related to the stored energy, but also to the temperature [34].

Figure 7 shows the distribution of the grain boundary and the misorientation angle of grains at different temperatures. The black color in the grain boundary figures represents the high-angle grain boundary (HAGB ≥ 15°), the green color represents the low-angle grain boundary (LAGB < 15°). It was observed that the grain boundary was very sensitive to the temperature at a high strain rate of 50% deformation. As the temperature increased to 1100 °C, the amount of low-angle grain boundaries decreased and the amount of high-angle grain boundaries increased, as shown in Figure 7a–d. The misorientation angle graphs show that the grain boundary frequency had a bimodal distribution at temperatures in the range of approximately 900 to 1200 °C, as shown in Figure 7e–h. This indicates that with the increase of temperature, some low-angle grain boundaries were transformed into high-angle grain boundaries to promote the dynamic recrystallization process. It is generally accepted that the DRX processes depend on the migration of the HAGB to eliminate the deformed structures. This agrees with the dynamic recrystallization at high temperatures, where the interface energy of the HAGB was higher and the migration of the grain boundary was promoted, which is the priority in the nucleation of dynamic recrystallization. Figure 7c,g shows that at 1100 °C, there were almost no low-angle grain boundaries, which indicated that complete dynamic recrystallization had occurred. As shown in Figure 7d, it also revealed that the grain rapidly grew at 1200 °C due to the high migration rate of the HAGB at a high temperature.

The proportions of the high-angle boundaries were 26% at 900 °C, 78% at 1000 °C, 93% at 1100 °C, and 58% at 1200 °C, as shown in Figure 8. Again, this illustrated the dynamic recrystallization process of the Inconel625 alloy at a high strain rate. As the temperature increased, the rate of dynamic recrystallization increased, the proportion of subgrain boundary (<2°) decreased, and the proportion of annealing twinning boundaries increased, as shown in Figure 8b. This was because as the temperature increased, the subgrain boundary gradually absorbed the lattice dislocations and gradually transformed into high-angle grain boundaries. At 1100 °C, the complete dynamic recrystallization occurred, the proportion of subgrain boundary decreased to 0.02, and the proportion of annealing twin boundaries reached 0.26. This indicated that the dynamic recrystallization process with an increase in temperature was accompanied by an increase in the annealing twin boundaries and a consumption of the subgrain boundaries. The change was opposite for the grain growth after the completion of dynamic recrystallization at 1200 °C, as shown in Figure 8b.

It is generally accepted that the DRX procedure is thermally activated [4]. As shown in Figure 6a–c, some fine grains were connected at the grain boundary and a so-called necklace structure formed. This is usually regarded as a typical feature of dynamic recrystallization. The recrystallized fraction map of the Inconel625 alloy at the strain rate of 10 s^−1^ under different temperatures is shown in Figure 9. The recrystallized fraction maps indicate the distribution of deformed and recrystallized grains, which was determined using HKL CHANNEL 5 software, where completely recrystallized grains are marked in blue, deformed grains are in red, and substructured grains (recrystallized grains with subgrains) are in yellow. Notably, the microstructure analysis was strongly sensitive to the deformation temperatures at a high strain rate. As shown in Figure 9, the volume fractions of the recrystallized grains were 17.3% at 900 °C, 65.8% at 1000 °C, 85.7% at 1100 °C, and 31.5% at 1200 °C. With increasing deformation temperature (1100 °C), numerous fine recrystallized and structured grains were observed. Deformed grains were hardly seen and an equiaxed grain structure was obtained, which indicated that complete dynamic recrystallization occurred. When deformed at 1200 °C, there were numerous grain structures observed. This indicated that the recrystallized grains grew at high temperatures after the completion of dynamic recrystallization, as shown in Figure 6e. There are some bulging boundaries observed in Figure 6a–c, where the part of the twinning boundaries crossing the internal grain intercepted the bulging boundary (Figure 6c). This observation, combined with Figure 8 and Figure 9, allows for the dynamic recrystallization mechanism of the Inconel625 alloy under a high strain rate to be obtained. The dynamic recrystallization mechanism of the Inconel625 alloy consisted of the continuous dynamic recrystallization of subgrain merging and rotation, whereas the discontinuous dynamic recrystallization with bulging grain boundaries induced by twinning was less dominant.

### 3.3. The Texture Evolution of the Inconel625 Alloy 

Figure 10 shows pole figures of the hot deformed microstructure of the Inconel625 alloy on the {100}, {110}, and {111} crystal faces at 900, 1000, 1100, and 1200 °C. The texture evolution was weakly correlated with the temperature. A random distribution was observed (Figure 10c), and the pole density was lowest, indicating that the Inconel625 alloy had favorable plasticity at 1100 °C. Compared to the cubic system {110}, {110}, and {111} in the standard pole figure, a strong shear texture {001}‹110› was observed in the {110} pole figure at 900 and 1000 °C. There were brass {110}‹1$\bar{1}$2› and annealing textures {$\bar{1}$11}‹1$\bar{1}$2› at 1100 and 1200 °C. To obtain a better understanding of the effect of the temperature on grain orientation, ODFs (orientation distribution figures) were obtained at different temperatures (Figure 11), and the Euler angles were calculated to obtain the texture at different temperatures. At 900 °C, due to the extremely uneven deformation, the texture shown here was mainly shear texture, and the intensity of the shear texture was very strong. The crystal orientation was random after increasing temperature, so the ODFs of the Inconel625 alloy, as shown in Figure 11b,c, showed no obvious texture. However, due to the completion of the dynamic recrystallization process, a strong brass texture {110}‹1$\bar{1}$2› was observed at 1200 °C, as shown in Figure 11d. 

The texture components of the Inconel625 alloy with a deviation of <15° at different temperatures were calculated using the CHANNEL 5 software, and the results are listed in Table 2. The dominant texture components were divided into two categories during compression: one represented the recrystallization textures of cube {001}‹100›, Goss {110}‹001›, and annealing {$\bar{1}$11}‹1$\bar{1}$2›; the other category represented the deformation textures of brass{$\bar{1}$01}‹1$\bar{2}$1›, S {123}‹63$\bar{4}$›, shear texture {001}‹110›, and copper {112}‹11$\bar{1}$›. With increasing temperature, the volume fraction of the recrystallization texture first increased, and then decreased. At 1100 °C, when the complete dynamic recrystallization occurred, some of the recrystallized grains grew and deviated from the original orientation, causing a decrease in the proportion of the recrystallization texture. At 1200 °C, the proportion of the recrystallization texture component was significantly lower. From 900 to 1200 °C, the proportion of the brass and S texture components increased from 1% to 10.6% and from 6% to 12.5% respectively, whereas the opposite trend was observed for the copper texture and shear texture. This may have been caused by the complex dynamic recrystallization process and the grain growth of the Inconel625 alloy at a high temperature and a high strain rate.

### 3.4. Discussion

The following is the discussion of the mechanism of the subgrain. It was indicated that the subgrain played a significant role during the DRX process. The measured kernel average misorientation (KAM) maps and local misorientation angle distributions of the Inconel625 alloy specimens after the high-strain rate deformation at high temperatures are given in Figure 12. The various colors in the KAM map demonstrate grains with different misorientation angles; the blue color in the KAM maps represents the subgrains (<2°). As shown in Figure 12a,b, some large strains appeared, owing to the formation of deformation bands during hot compression and incomplete dynamic recrystallization caused by deformation incompatibility at 900 and 1000 °C. A large number of dislocations were accumulated at the grain boundaries, providing conditions for the nucleation of subgrains and the DRX at grain boundaries. As shown in Figure 12d, a small quantity of large strains and the accumulation density area appeared, which may have been caused by complex grain growth after the completion of DRX. As shown in Figure 12e–h, the average angle initially decreased, then slightly increased. The change was identical for the geometrically necessary dislocation (*GND*) density.

The proportions of the subgrains were 0.54 at 900 °C, 1.61 at 1000 °C, 3.76 at 1100 °C, and 1.08 at 1200 °C, as shown in Figure 13a. As the temperature increased, the number of subgrains first increased and then decreased. The *GND* density was calculated by Equation (3), as shown in Figure 13b. The opposite trend of subgrain evolution was observed for the *GND* density. It indicated that the nucleation of subgrains was dependent on the annihilation of the dislocations during the DRX process. At 1100 °C, the average local misorientation angle and the *GND* density were small after fully dynamic recrystallization. At a temperature of 1200 °C, the DRX was almost completed and the subgrains grew by mutual annexation, resulting in the decrease of subgrain and the increase of *GND* density. Calculation for the *GND* density was obtained from the following:(3)$$\rho^{GND}=\frac{2\theta }{ub},$$
where $\rho^{GND}$ is the *GND* density, θ is the local misorientation average angle, *u* is the step length (100 nm), and *b* is the Burgers vector (0.254 nm for the Inconel625 alloy).

## 4. Conclusions

In this study, the dynamic recrystallization behavior of the Inconel625 alloy was prepared by the hot compression and EBSD analytic techniques. The texture evolution and DRX mechanism were observed and analyzed. The outcomes of this work are summarized as follows:The true stress–true strain curves exhibited a clear steady state flow rule at different temperatures and different strain rates. The adiabatic heating effect decreased with decreasing temperature and increasing strain rate.The dynamic recrystallization process with increasing temperature was accompanied by an increase in twinning boundaries and a consumption of the subgrains, while the change of subgrains and twinnings was opposite for the grain growth after the completion of dynamic recrystallization at 1200 °C. There were almost no low-angle grain boundaries at 1100 °C, which indicated that complete dynamic recrystallization had occurred.At a high strain rate, the continuous dynamic recrystallization with subgrain amalgamation and rotation of the Inconel625 alloy played a significant role, whereas the discontinuous dynamic recrystallization, with a bulging boundary induced by twinning, played an auxiliary role.At a high strain rate, the texture was weakly correlated with the change in temperature. The crystal orientation was random after temperature increase, and the Inconel625 alloy showed no obvious texture. The subgrain played a significant role during the DRX process. The nucleation of subgrain and DRX was dependent on the motion of the dislocations during the hot deformation.

## Figures and Tables

**Figure 1 materials-12-00510-f001:**
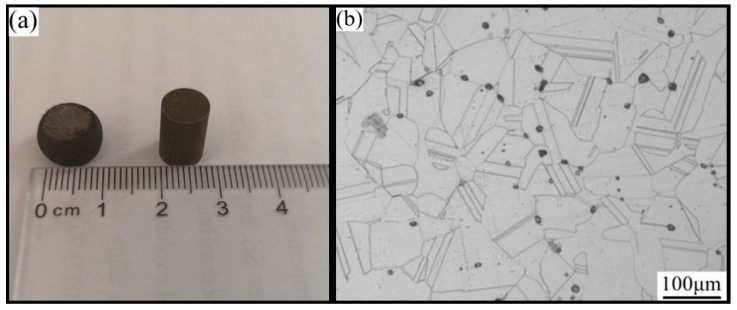
The Inconel625 alloy specimens and original microstructure. (**a**) Specimens (**b**) Original microstructure.

**Figure 2 materials-12-00510-f002:**
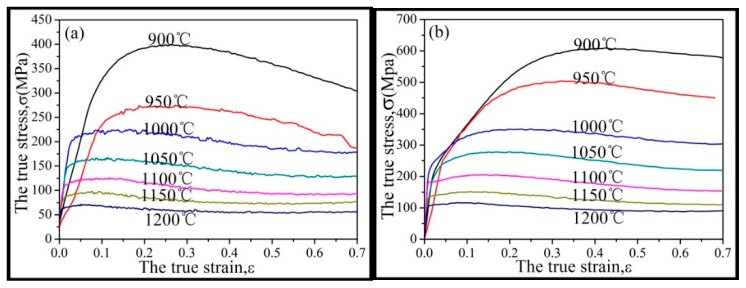
True stress–strain curves and peak stress curves of Inconel625 alloy at different strain rates and temperatures. (**a**) $\dot{\varepsilon }$ = 0.01 s^−1^; (**b**) $\dot{\varepsilon }$ = 0.1 s^−1^; (**c**) $\dot{\varepsilon }$ = 1 s^−1^; (**d**) $\dot{\varepsilon }$ = 10 s^−1^; (**e**) The peak stress curves.

**Figure 3 materials-12-00510-f003:**
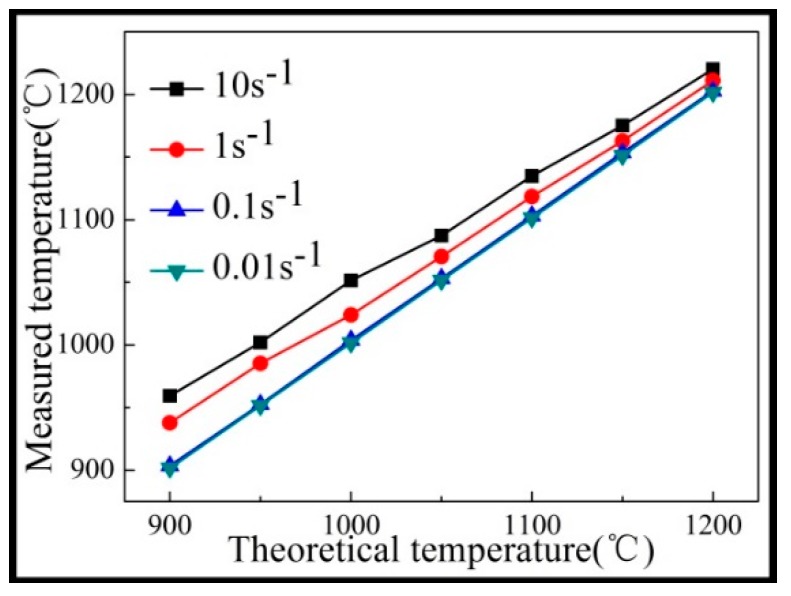
Adiabatic heating effect of the Inconel625 alloy during hot compression.

**Figure 4 materials-12-00510-f004:**
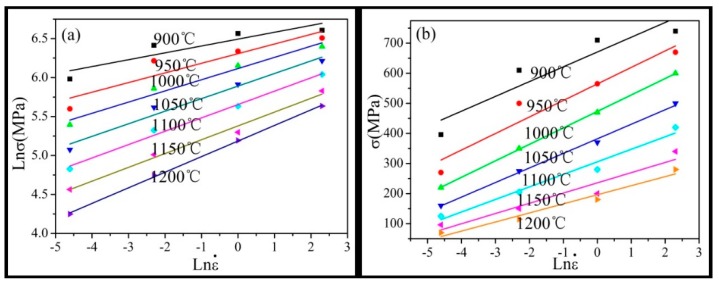
Relationship between peak stress and strain rate. (**a**) Inσ −$\;\mathrm{In}\;\dot{\varepsilon }$; (**b**) σ −$\;\mathrm{In}\dot{\varepsilon }$.

**Figure 5 materials-12-00510-f005:**
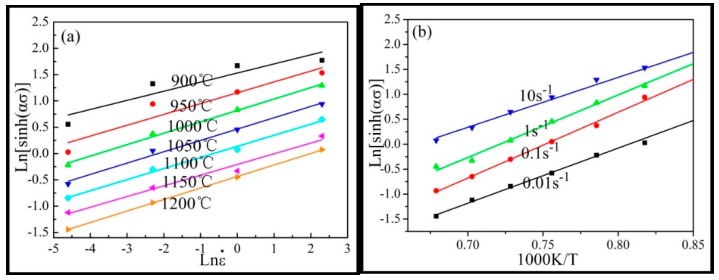
Relationship between In[sinh(ασ)] and $\mathrm{In}\;\dot{\varepsilon }$ and T^−1^. (**a**) In[sinh(ασ)] −$\;\mathrm{In}\;\dot{\varepsilon }$; (**b**) In[sinh(ασ)] − T^−1^.

**Figure 6 materials-12-00510-f006:**
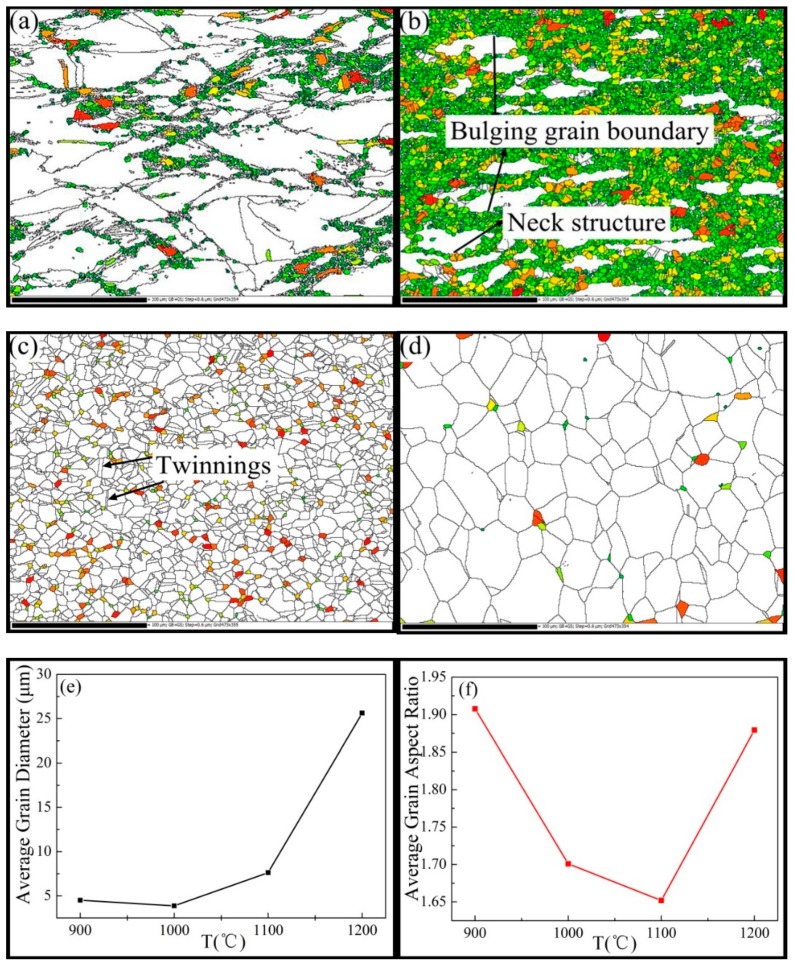
Grain size and morphology of Inconel625 at different temperatures. (**a**) 900 °C; (**b**) 1000 °C; (**c**) 1100 °C; (**d**) 1200 °C; (**e**)Average Grain Diameter; (**f**)Average Grain Aspect Ratio.

**Figure 7 materials-12-00510-f007:**
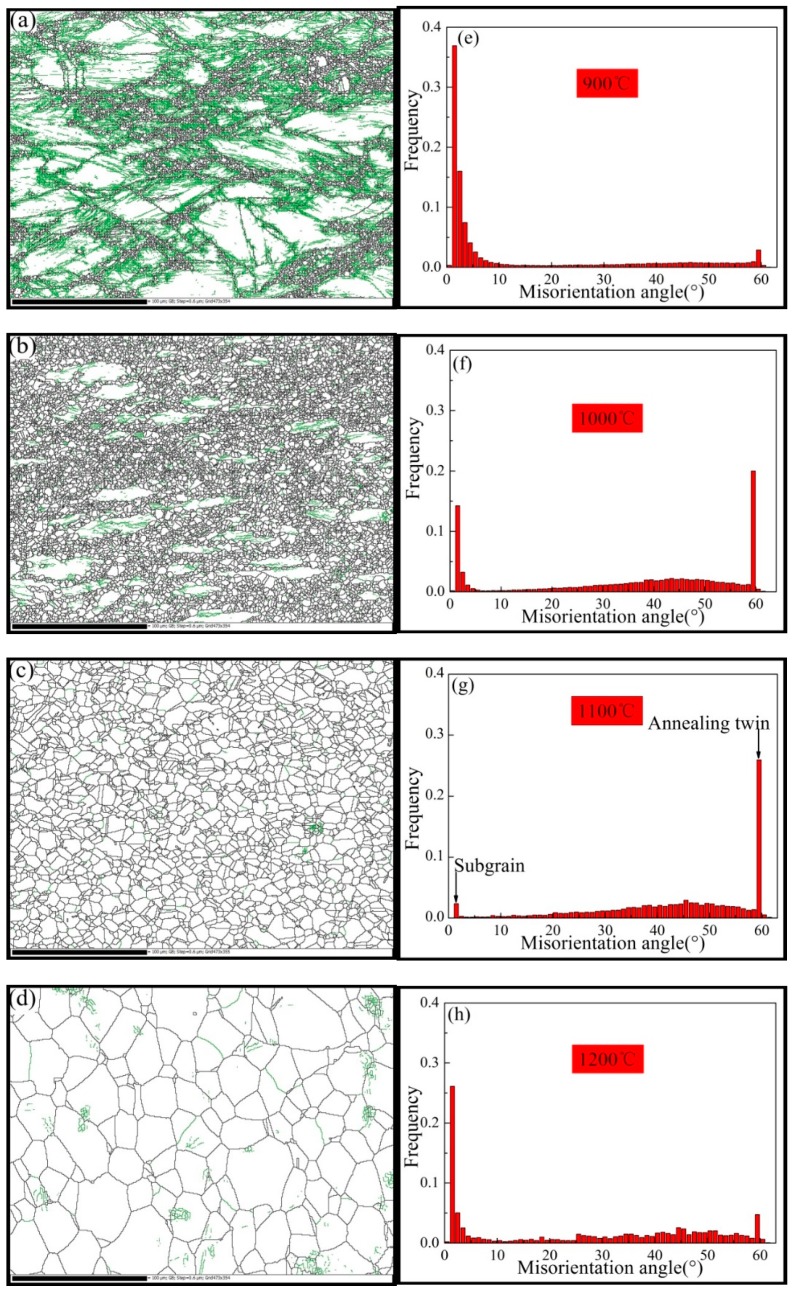
Grain boundary distributions and the correlated grain misorientation angles at different temperatures. (**a,e**) 900 °C; (**b,f**) 1000 °C; (**c,g**) 1100 °C; (**d,h**) 1200 °C.

**Figure 8 materials-12-00510-f008:**
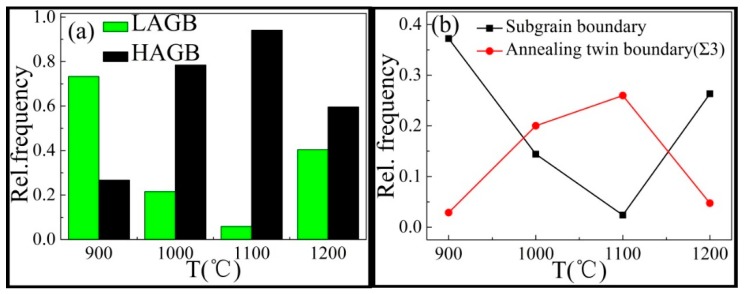
The frequency of the low-angle grain boundary (LAGB), high-angle grain boundary (HAGB), subgrain boundary (SB), and annealing twin boundary (Σ3) at different temperatures. (**a**) The LAGB and the HAGB; (**b**) The SB and the Σ3.

**Figure 9 materials-12-00510-f009:**
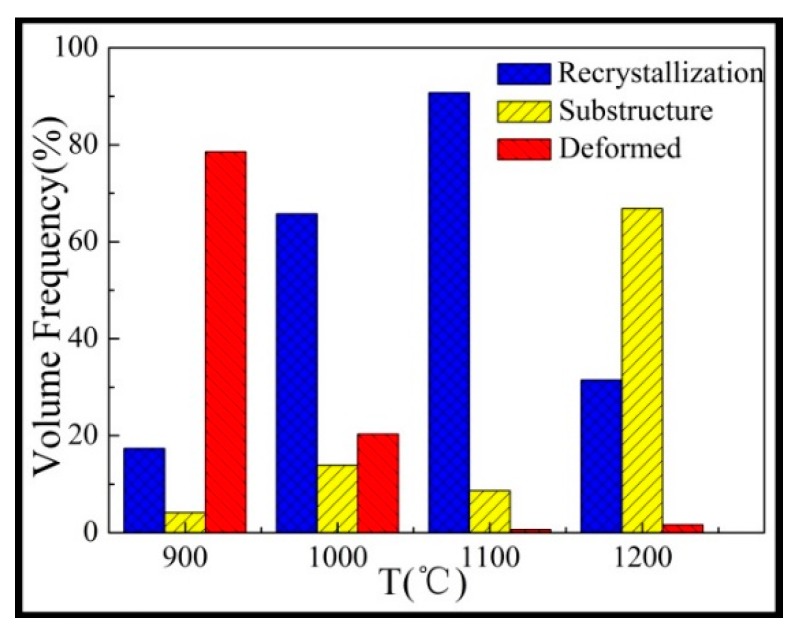
The volume fraction of the recrystallization grain.

**Figure 10 materials-12-00510-f010:**
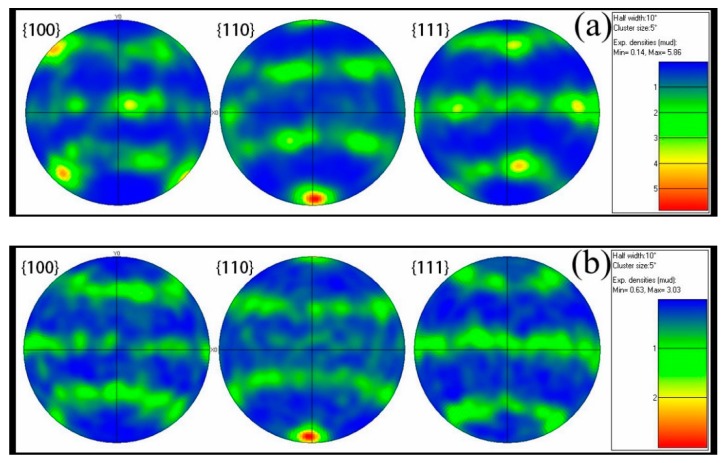
The pole figures of the Inconel625 alloy at different temperatures. (**a**) 900 °C; (**b**) 1000 °C; (**c**) 1100 °C; (**d**) 1200 °C.

**Figure 11 materials-12-00510-f011:**
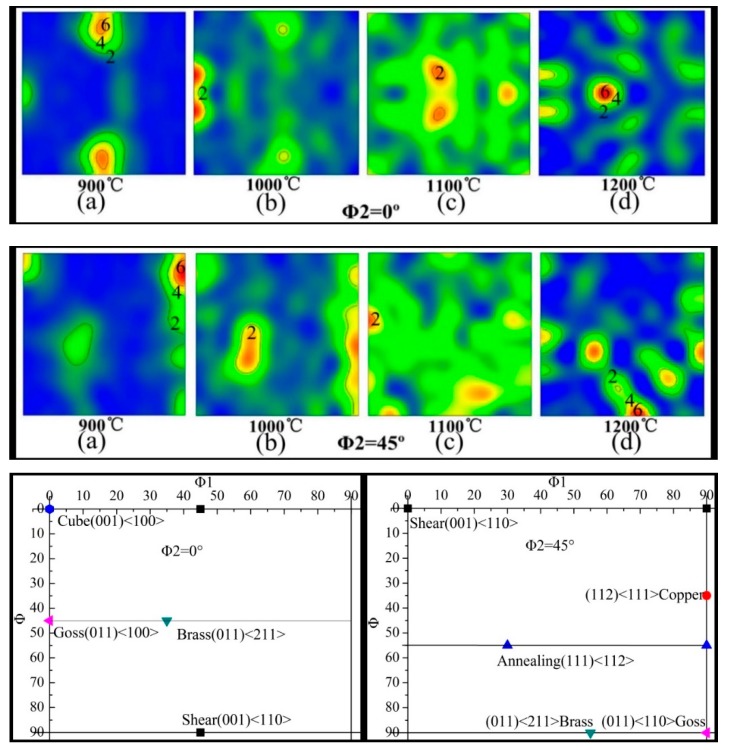
The orientation distribution figures (ODFs) of Inconel625 alloy at different temperatures. (**a**) 900 °C; (**b**) 1000 °C; (**c**) 1100 °C; (**d**) 1200 °C.

**Figure 12 materials-12-00510-f012:**
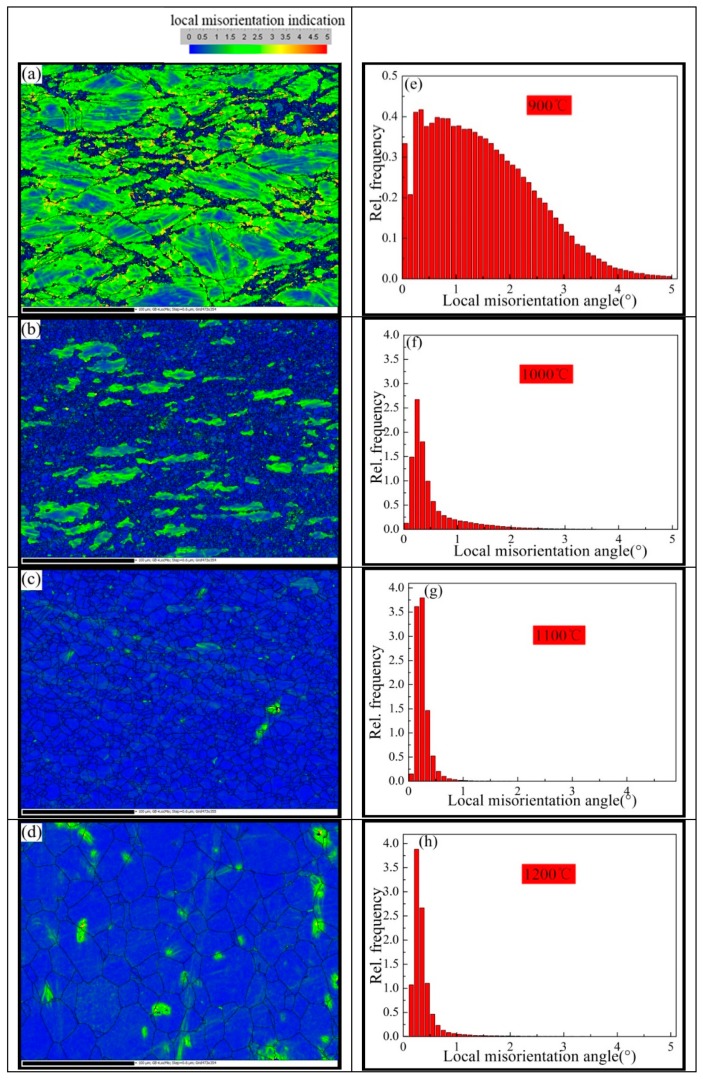
Kernel average misorientation (KAM) maps and local misorientation angles at different temperatures. (**a,e**) 900 °C; (**b,f**) 1000 °C; (**c,g**) 1100 °C; (**d**,**h**) 1200 °C.

**Figure 13 materials-12-00510-f013:**
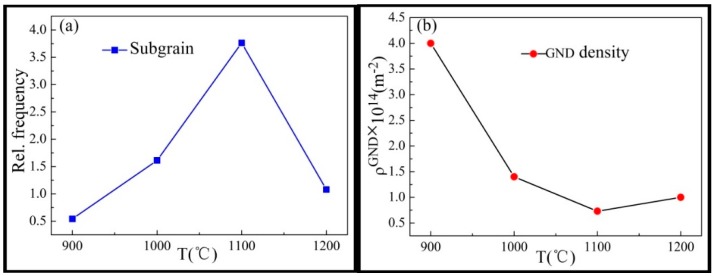
The frequencies of subgrains and geometrically necessary dislocation (*GND*) densities measured at different temperatures. (**a**) The frequencies of subgrains; (**b**) *GND* densities.

**Table 1 materials-12-00510-t001:** The major chemical compositions of the Inconel625 (mass percent, wt %).

C	Cr	Ni	Co	Mo	Al	Fe	Ti	Nb	Si	Mn
0.042	21.77	60.63	0.19	8.79	0.21	3.68	0.40	3.75	0.12	0.2

**Table 2 materials-12-00510-t002:** Texture compoents (%) for Inconel625 alloy under different temperatures.

Condition/°C	Cube	Goss	Annealing	Brass	S	Copper	Shear	Random	Total Recrystallization Texture
900	0.429	4.01	8.37	0.977	5.9	9.95	10.5	57.862	12.809
1000	1.6	4.42	7.98	3.65	6.78	6.02	3.56	65.99	14
1100	2.05	1.66	4.45	6.03	9.73	3.93	1.71	70.44	8.16
1200	0.427	3.26	2.72	10.6	12.5	1.81	5.24	63.623	6.407

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
