# Peer review of "Study of the Dynamic Recrystallization Process of the Inconel625 Alloy at a High Strain Rate"

_materials, 2019, doi:10.3390/ma12030510_

Round 1

Reviewer 1 Report

Overall, the manuscript offers some interesting insights regarding the hot deformation behavior at different strain rates and temperatures for the GH3625 superalloy (which is widely used in aerospace, nuclear power, shipbuilding and other industrial fields). It is an interesting research and the reviewer suggests to accept this paper for publication in Materials, but only after a major revision to cover the following comments and suggestions.

Although the subject under discussion is interesting and the submission contains valuable publishable content, this reviewer considers that the article has some serious flaws (the paper is not well written and has structural and presentation problems), not being acceptable for publication in Materials in its current form. I recommend a rigorous article restoration considering also the appropriate template provided for editing the manuscript.

Throughout the manuscript English is not at the level required for a scientific publication. A further proof read of the English is required (language: spelling, style and grammar). The writing has to be clear, concise and interesting. Please improve writing! The reviewer recommends and considers that this research work needs to be reviewed by a native English speaker as well. Some scientific terms are not used correctly. Maybe it's about the misuse of some terms in English, but please be very careful with the terminology or explain better the aspects that are not consecrated.

The title of the paper can be improved in order to properly reflect the content of the paper.

The abstract can be improved. The abstract has to reflect the content of the paper accurately and has to bring out the main points of the paper. Abstract has to be specific and representative and the motive for the research has to be indicated. It should state concisely the goals, methods, principal results and major conclusions of the paper. Please improve!

Abstract, l. 21 – Please correct “completes” to “complete”.

Abstract, l. 20-21 – Please consider rephrasing the sentence: “At 1200 , the dynamic recrystallization was almost completes, the grains had merged.”

l. 26, Keywords – Please provide at least one more keyword and delete “The” from “The GH3625 alloy”.

The introduction / literature survey is too brief and the contribution of the paper to the state of the art is not so clearly presented. A better, more thorough state of the art is required. Also, in the Introduction, the references are used in large groups. Please avoid that, disseminating the references by sentences or ideas.

l. 30-31, Introduction – Please rephrase the sentence: “At present, pipes and bars are used in the processing and production of nickel-based superalloy in various fields.”

l. 33 – Please replace “large” with “high”.

l. 34 – Please correct “difficult” to “difficulties”.

l. 34-35 – Combine the two paragraphs (remove the “Enter” after “compression”).

l. 36 – Please replace “microstructure analysis” with “to analyze the microstructure”.

l. 39 – Please correct “the microstructure analysis and hot deformation behavior” to “the microstructure and the hot deformation behavior”.

l. 46 – Correct “GH6325” to “GH3625”.

The experimental methods section needs to be strengthened by including all major and minor details as well as the logic behind the selection of experimental parameters, should the work to be replicated at a later date. Any comprehensive study includes also an explanation of the drawbacks of experimental techniques. Please improve section 2 by giving complete details on all performed experiments and investigations! Also, please mention all the equipment and devices used, giving also more details about them (manufacturer, model, company location – city, state, country).

l. 49-50 – Please correct and consider rephrasing the sentence: “The material was a solid solution strengthened nickel-based deformation superalloy with the chemical compositions in table 1.”

l. 50 – Replace “is” with “were”.

l. 51 – The specimens were obtained from a “cylindrical extruded tube”??? Tube? Maybe it was a cylindrical extruded bar?

l. 54-55 – Please rephrase the sentence: “After the hot compression tests, the specimens was determined using electron backscatter diffraction (EBSD).”????

In Table 1, are you sure about the chemical symbol “Mu”? Mu is the symbol for Muonium.

Figure 1 – Please improve the quality of the figures by using a smaller font and by explaining σ and ε on the two axis (I think is better to write “True stress, σ [MPa]” and “True strain, ε” instead of simply σ and ε). Also, it would be better to insert a legend for each figure to explain the curves, in order not to overload the graphs.

Figure 1e – What does “Peak stress” stands for and how do you explain the values from the y axis in Figure 1e (you have on this axis values ranging from -1 to 3 ????) You made some compression tests. Maybe it would be better to refer to the “ultimate compression strength” or at least to the “stress at peak” instead??? And again, how can those values be explained???

l. 87 – “(C)” – Please remove Caps.

Figure 2 – y axis – Please correct the typo.

Figure 2 – Please use a smaller font. This observation is valid for almost all the graphs in the manuscript.

l. 99-108 – Please use the lowercase letter “l” (as in “Lima”) for “ln” and not the capital/uppercase letter “I” (as in “India”).

Please explain better and give more details about the graphs from Figures 3 and 4 and also about the constitutive equation and its calculation. A discussion on those relationships and graphs is required.

l. 109 – Please reconsider or rephrase this subheading.

l. 111 – “at different temperatures”? I think that it would be more correct “after compression at different temperatures”. Same observation for line 133.

l. 110-112 – Please consider rephrasing.

l. 116 – Please correct “recrystallized” to “recrystallization”.

l. 137, l. 138 – “the amount of……”. Please correct!

l. 152 – “dynamics” – Please correct to “dynamic”.

l. 168-169 – Please consider rephrasing!

l. 199 – Please edit the figure caption according to the template and consider rephrasing. Same observation for line 203.

l. 204 – Please correct the typos!

l. 208 – Please explain all abbreviations the first time they appear in the text (DRX)!

l. 209 – Please delete “before”!

l. 215 – Please correct “increasing” to “increases”, “decreased” to “decreases”.

l. 215 – Please replace “then slightly increased” with “and then is slightly increased”.

l. 219 and equation 2 – It would be better to use ρGND instead of ρCND for the GND density?

l. 220 – “Burr vector”??? Maybe is “Burgers vector”?

l. 220 – Please use the same designation for the investigated material in all the manuscript. Please replace “In625” with “GH3625”! Or, as an alternative you can use the IN625 notation in all your paper instead of the Chinese designation GH3625 (and I think it would be better this way).

l. 257-258 – Please explain also Fig. 10 (e). Please edit the figure caption according to the template!

The conclusions have to be accurate and supported by the content in a logic order. Please improve conclusions of your paper!

l. 259 – Correct “Conclusion” to “Conclusions”.

l. 260-262 – Please consider rephrasing the sentence!

l. 265-266 – Use the same font size please! Replace “increases in the” with “increasing”!

l. 279 – Correct “Reference” to “References”!

Please edit the references list according to the required template, giving also all the specified details for each paper. Remove duplicate numbering of references (1. [1]…., 2. [2]…. etc.)

Author Response

Dear reviewer,

Thank you for your letter and for the reviewers’ comments concerning our manuscript. Those comments are all valuable and very helpful for revising and improving our paper, as well as the important guiding significance to our researches. We have studied the comments carefully and have made revision in the paper. These changes will not influence the content and framework of the paper. The detailed corrections are listed below point by point:

Point 1: Throughout the manuscript English is not at the level required for a scientific publication. A further proof read of the English is required (language: spelling, style and grammar). The writing has to be clear, concise and interesting. Please improve writing! The reviewer recommends and considers that this research work needs to be reviewed by a native English speaker as well. Some scientific terms are not used correctly. Maybe it's about the misuse of some terms in English, but please be very careful with the terminology or explain better the aspects that are not consecrated.

Response 1: We have revised the whole manuscript carefully and tried to avoid any grammar or syntax error. In addition, we have invited a professor to check the English, who is a proficient English speaker. We believe that the language is now acceptable for the review process.

Point 2: The title of the paper can be improved in order to properly reflect the content of the paper. The abstract can be improved. The abstract has to reflect the content of the paper accurately and has to bring out the main points of the paper. Abstract has to be specific and representative and the motive for the research has to be indicated. It should state concisely the goals, methods, principal results and major conclusions of the paper. Please improve!

Response 2: We have revised title and abstract according to the reviewer’s suggestion. We state concisely the goals, methods, principal results and major conclusions of the paper.

Point 3: The introduction / literature survey is too brief and the contribution of the paper to the state of the art is not so clearly presented. A better, more thorough state of the art is required. Also, in the Introduction, the references are used in large groups. Please avoid that, disseminating the references by sentences or ideas.

Response 3: We have revised introduction according to the reviewer’s suggestion. We describe the previous study related to the Ni-based superalloy.

Point 4: The experimental methods section needs to be strengthened by including all major and minor details as well as the logic behind the selection of experimental parameters, should the work to be replicated at a later date. Any comprehensive study includes also an explanation of the drawbacks of experimental techniques. Please improve section 2 by giving complete details on all performed experiments and investigations! Also, please mention all the equipment and devices used, giving also more details about them (manufacturer, model, company location – city, state, country).

Response 4: We have strengthened the experimental methods section according to the reviewer’s suggestion. We describe in details on all performed experiments and investigations.

Point 5: Although the subject under discussion is interesting and the submission contains valuable publishable content, this reviewer considers that the article has some serious flaws (the paper is not well written and has structural and presentation problems), not being acceptable for publication in Materials in its current form. I recommend a rigorous article restoration considering also the appropriate template provided for editing the manuscript.

Reviewer 2 Report

The paper contains interesting results about deformation of GH3625 superalloy at high temperatures. The paper needs following improvement:

The paper is missing the description of the initial microstructure and phase composition. Following details are required to understand the content well: room temperature SEM micrograph) (or optical micrograph), EBSD map at room temperature and also XRD pattern, if possible.

The language contains a lot of grammar errors, e.g. "After the hot compression tests, the specimens was (were) determined using electron backscatter diffraction (EBSD)." or "Figure 9. KAM maps, subgrains and GND density measured in (at) the different temperatures". From this figure description it seems that the measurement was done at the temperature, but in reality it was after hot deformation at that temperature.

There are also senteces which make no sense, e.g. "The constitutive equation of the GH3625 alloy is obtained using Arrhenius constitutive equation:" or "The grain boundary has a ragged edge due to the stress concentration, which facilitates the dynamic recrystallized and nucleation of the grains near these short grain boundaries, as shown in Fig. 5(a)."

Author Response

Dear reviewer,

Thank you for your letter and for the reviewers’ comments concerning our manuscript. Those comments are all valuable and very helpful for revising and improving our paper, as well as the important guiding significance to our researches. We have studied the comments carefully and have made revision in the paper. These changes will not influence the content and framework of the paper. The detailed corrections are listed below point by point:

Point 1: The paper is missing the description of the initial microstructure and phase composition. Following details are required to understand the content well: room temperature SEM micrograph) (or optical micrograph), EBSD map at room temperature and also XRD pattern, if possible.    

Response 1: We have descripted the initial microstructure in the section of experimental methods. We have listed the optical micrograph of experimental material at room temperature.

Point 2: The language contains a lot of grammar errors, e.g. "After the hot compression tests, the specimens was (were) determined using electron backscatter diffraction (EBSD)." or "Figure 9. KAM maps, subgrains and GND density measured in (at) the different temperatures". From this figure description it seems that the measurement was done at the temperature, but in reality it was after hot deformation at that temperature.

Response 2: According to the reviewer’s suggestion, we revised in details about words, sentences, grammar, and the caption of figures and tables.

Point 3: There are also sentences which make no sense, e.g. "The constitutive equation of the GH3625 alloy is obtained using Arrhenius constitutive equation:" or "The grain boundary has a ragged edge due to the stress concentration, which facilitates the dynamic recrystallized and nucleation of the grains near these short grain boundaries, as shown in Fig. 5(a).".

Response 3: We have revised these sentences according to the reviewer’s suggestion.

Reviewer 3 Report

This paper presents results of investigations on development of microstructure after hot deformation processes at different temperature range of 900-1200oC. Microstructure analysis were carried out by means of EBSD technique. The manuscript is chaotically prepared, especially a discussion concerning particular Figures presented in the text. Before acceptance of the manuscript for printing Authors should take into account the following comments:

1.      Spaces between numerical value and “oC” should be delated (e.g. pp. 2, line 52, 53).

2.      All abbreviations of "Fig." should be extended to full form "Figure". Moreover, all of descriptions pictures and tables should be formatted according to the mdpi website section -> "Materials — Instructions for Authors".

3.      The Figure 1: values of “peak stress” axis are incorrect.

4.      The Figure 2: there is a mistake on axis “measured temperatuyre” – should be “measured temperature”.

5.      There are three different formats of sign “-“ – it should be unified.

6.      The Figure 5: scale bars are invisible. The Figure 5 e and f have to be presented as a separated, additional Figure.

7.      The Figure 6 a, b and d are not indicated in the text. The Figure 6 e and f have to be presented as a separated, additional Figure. Moreover, current marks of particular Figures (a, b, c and d) are incorrect, because there are eight Figures and only four marks – it have to be changed.

8.      The Figure 7 a, b and d and the Figure 8 a are not indicated in the text.

9.      The Figure 9 – see comments concerning Figure 6.

10.  The Figure 10 c-e are not indicated in the text correctly.

11.  The paper does not have any discussion. There are not any reference to literature sources in the text in the “Results and discussion” section.

12.  A sentence “Taking all these data into analysis, this paper may safely come to the conclusion that the hot deformation behavior and microstructure analysis of the GH3625 alloy at different temperatures and a high strain rate by using the high-resolution EBSD technique.” – what does it mean? (pp. 13, lines: 260-262).

13.  Conclusions should be numerated.

14.  There is some kind of mistake in a font of the first conclusion.

15.  The References should be formatted according with website section -> "Materials — Instructions for Authors".

Author Response

Dear reviewer,

Thank you for your letter and for the reviewers’ comments concerning our manuscript. Those comments are all valuable and very helpful for revising and improving our paper, as well as the important guiding significance to our researches. We have studied the comments carefully and have made revision in the paper. These changes will not influence the content and framework of the paper. The detailed corrections are listed below point by point:

Point 1: This paper presents results of investigations on development of microstructure after hot deformation processes at different temperature range of 900-1200oC. Microstructure analysis were carried out by means of EBSD technique. The manuscript is chaotically prepared, especially a discussion concerning particular Figures presented in the text.

Response 1: We have revised the structural and presentation of this manuscript. According to the reviewer’s suggestion, we revised in details about words, sentences, and the caption of figures and tables.

Point 2: A sentence “Taking all these data into analysis, this paper may safely come to the conclusion that the hot deformation behavior and microstructure analysis of the GH3625 alloy at different temperatures and a high strain rate by using the high-resolution EBSD technique.” – what does it mean? (pp. 13, lines: 260-262).

Response 2: We have revised this sentence in this manuscript. In this study, the dynamic recrystallization behavior of the In625 alloy has been prepared by the hot compression and EBSD analytic technique. The texture evolution and DRX mechanism have been observed and analysed. The outcomes of this work could be summarized as follows:

Point 3:  Conclusions should be numerated. There is some kind of mistake in a font of the first conclusion. The References should be formatted according with website section -> "Materials — Instructions for Authors".

Response 3: We have revised conclusions and references according to the reviewer’s suggestion.

Reviewer 4 Report

 The authors have investigated the impact of temperature and strain rate on the hot deformation behavior and microstructure of GH3625 superalloy. GH3625 is a Ni-based superalloy with several practical applications and hence the alloy is an interesting system for study. In this manuscript the authors have presented many new results, which probably deserves to be published at some stage. However, I have some major concerns regarding the manuscript and I take only take a decision depending on the response by the authors.

(1) Hot deformation process has its own difficulties. Previous studies on cold deformation of this alloy are available (Transactions of Materials and Heat Treatment 38(2):178-184). So, the motivation behind the present investigations is not entirely clear. Can the authors define clearly a “key-goal” which they wanted to achieve in their study?

(2) Similarly, in the conclusions the authors have summarized their observations. However, it is unclear how these observations improve the practical applicability or even solve some of the known limitations of this alloy.

(3) In Section 2, can the authors comment on the method used for determining chemical composition?

(4) In sub-section, authors have separated the stress-strain curve into work hardening stage and dynamic softening stage. How did the authors classify these stages? Is it merely based on the shape of the stress-strain curve? In the same discussion, the authors mentioned, “…increase in the dislocation density up to a certain critical value…” Is this critical value of dislocation density system dependent? Which factors determine the critical density?

(5) In Figure 5, sub-section 3.2, authors describe recrystallization of grains. What is the driving force for the grain growth? Is the strain-rate 50%? How is this strain rate connected with the strain rates of Figure1 and Figure2? The average grain diameter (Figure 5(e)) increases almost exponentially.  Can the authors comment on the origin of this exponential behavior? The colour code and corresponding length scale used in Figure 5 is not defined properly.

(6) In the same subsection (3.2), the authors mentioned, “…interface energy of the high-angle grain boundary is higher…” Can the authors explain the reason behind this? What is tentatively the range of the mentioned interface energy?

(7) Editing of language at some places is required.

Author Response

Dear reviewer,

Thank you for your letter and for the reviewers’ comments concerning our manuscript. Those comments are all valuable and very helpful for revising and improving our paper, as well as the important guiding significance to our researches. We have studied the comments carefully and have made revision in the paper. These changes will not influence the content and framework of the paper. The detailed corrections are listed below point by point:

Point 1: (1) Hot deformation process has its own difficulties. Previous studies on cold deformation of this alloy are available (Transactions of Materials and Heat Treatment 38(2):178-184). So, the motivation behind the present investigations is not entirely clear. Can the authors define clearly a “key-goal” which they wanted to achieve in their study?

Response 1: The key-goal of this manuscript is the dynamic recrystallization (DRX) behavior and texture evolution of In625 alloy during the hot deformation.

Point 2: (2) Similarly, in the conclusions the authors have summarized their observations. However, it is unclear how these observations improve the practical applicability or even solve some of the known limitations of this alloy.

Response 2: We hope to optimize the hot working technology of nickel based superalloys by studying the microstructure evolution of In625 alloy during hot deformation, and solve the problems in the processing of high-temperature alloy products, for example, the problem of tube cracking of the In625 alloy under high-speed hot extrusion process.

Point 3: (3) In Section 2, can the authors comment on the method used for determining chemical composition?

Response 3: The chemical compositions were measured by Electron Microprobe (EPMA, Lanzhou, China).

Point 4: (4) In sub-section, authors have separated the stress-strain curve into work hardening stage and dynamic softening stage. How did the authors classify these stages? Is it merely based on the shape of the stress-strain curve? In the same discussion, the authors mentioned, “…increase in the dislocation density up to a certain critical value…” Is this critical value of dislocation density system dependent? Which factors determine the critical density?

Response 4: The separation of the true stress-strain curve based on the peak stress and shape of the stress-strain curve. The correlation between this critical value and dislocation density system, and the factors are for further study.

Point 5: In Figure 5, sub-section 3.2, authors describe recrystallization of grains. What is the driving force for the grain growth? Is the strain-rate 50%? How is this strain rate connected with the strain rates of Figure1 and Figure2? The average grain diameter (Figure 5(e)) increases almost exponentially.  Can the authors comment on the origin of this exponential behavior? The colour code and corresponding length scale used in Figure 5 is not defined properly.

Response 5: The growth of the grains is realized by the motion of grain boundary, which is not only related to the stored energy but also to the temperatures. This study focused on the temperature.

Point 5: (6) In the same subsection (3.2), the authors mentioned, “…interface energy of the high-angle grain boundary is higher…” Can the authors explain the reason behind this? What is tentatively the range of the mentioned interface energy?

Response 5: In the dynamic recrystallization process, it is generally involved in the migration of high-angle grain boundaries to eliminate the deformed structure, rather than the low-angle grain boundaries. Thus, it can be considered that the interface energy of the high-angle grain boundary is larger than that of the low-angle grain boundary because the high-angle grain boundary is more unstable from the thermodynamic point of view during the recrystallization process. The range of the mentioned interface energy needs further study.

Point 5: (7) Editing of language at some places is required.

Response 5: We have revised the whole manuscript carefully and tried to avoid any grammar or syntax error. In addition, we have invited a professor to check the English, who is a proficient English speaker. We believe that the language is now acceptable for the review process.

Round 2

Reviewer 1 Report

The manuscript was improved in light of the original review and now warrants publication in Materials.

Kind regards!

Reviewer 2 Report

The paper was significantly improved.

Reviewer 3 Report

In the current version there are changes which are not indicated by red font, even the title – this is confusing. Some comments have not been considered and Authors did not explain why, in the section “Author response”.

1.      All “figures” and “tables” should be written from capital letters, in the text.

2.      The descriptions of pictures and tables have been improved, but still have to be formatted according to the mdpi website section -> "Materials — Instructions for Authors".

3.      The Figure 6 and 7: scales are invisible. The Figure 6 e and f have to be presented as a separated, additional Figure.

4.      The Figure 8 a are not indicated in the text.

5.      Spaces between numerical value and “oC” should be delated.

6.      The References have been changed, but still need a formatting according with website section -> "Materials — Instructions for Authors".

Reviewer 4 Report

The authors have tried to address most of my comments. Although there is still scope for further improvement, I find their response acceptable. Based on their response letter and the improved version of the manuscript, I believe the current version of the manuscript can now be published.